# Intraperitoneal BromAc^®^ Does Not Interfere with the Healing of Colon Anastomosis

**DOI:** 10.3390/cancers15133321

**Published:** 2023-06-24

**Authors:** Ahmed H. Mekkawy, Mohammad Breakeit, Krishna Pillai, Samina Badar, Javed Akhter, Sarah J. Valle, David L. Morris

**Affiliations:** 1Mucpharm Pty Ltd., Sydney, NSW 2217, Australia; ahmed@mucpharm.com (A.H.M.); mohammad.breakeit@health.nsw.gov.au (M.B.); panthera6444@yahoo.com.au (K.P.); javed@mucpharm.com (J.A.); sarah@mucpharm.com (S.J.V.); 2Department of Surgery, St George Hospital, Sydney, NSW 2217, Australia; samina.badar@unsw.edu.au; 3St George & Sutherland Clinical School, University of New South Wales, Sydney, NSW 2217, Australia; 4Intensive Care Unit, St George Hospital, Sydney, NSW 2217, Australia

**Keywords:** acetylcysteine, BromAc^®^, bromelain, colon anastomosis, rat model

## Abstract

**Simple Summary:**

The drug BromAc^®^ has been used in the treatment of pseudomyxoma peritonei, a rare peritoneal cancer. Often, patients with pseudomyxoma peritonei require colon resection with reconnection. Hence, we investigated whether BromAc^®^ affects the healing of bowel wounds using a rat model. After colon-resection surgery, BromAc^®^ was administered to a group of rats. Another group (control) of rats received saline. The colons were then left to heal for different time intervals. Next, the colons were assessed for healing using the burst-pressure test, whilst the effect of the BromAc^®^ on the internal organs was investigated with histology. The results showed no significant differences in the healing of the colon wounds between the BromAc^®^-treated and the control rats. In addition, there was no difference in internal organ histology between the BromAc^®^-treated rats and the controls. Hence, the administration of intraperitoneal BromAc^®^ is safe for the healing of colon resection.

**Abstract:**

A combination of bromelain and acetylcysteine, BromAc^®^, is an efficient intraperitoneal mucolytic for thick mucus secreted in pseudomyxoma peritonei (PMP). Patients with PMP quite often undergo colon anastomosis. Hence, we investigated the effect of the intraperitoneal delivery of BromAc^®^ on colon-anastomosis healing in a rat model. Sixteen Wistar rats were divided into two groups (N = 8). The controls received intraperitoneal saline after anastomosis, whilst the other group received BromAc^®^. They were monitored for body-weight and general health parameters. Half the rats in each group (N = 4) were culled at 4 or 13 days post-surgery for assessment. The healing process of the tissues was assessed by burst pressure and collagen density with histology to assess the integrity of the internal organs. The results indicated that there was a similar pattern of weight fluctuation during the experiment, although the rats treated with the BromAc^®^ showed slightly greater weight loss during the first 4 days. Although the burst pressure was similar in both groups, the BromAc^®^ group at day 13 showed a slightly higher burst pressure, which was complemented by a higher collagen density (albeit not statistically significant). The histology of the internal organs was comparable to those of the controls. This study indicates that the intraperitoneal delivery of BromAc^®^ in a rat model does not interfere with the healing process of colonic anastomosis.

## 1. Introduction

Pseudomyxoma peritonei (PMP) is a rare cancer that primarily originates in the appendix; however, it can also originate in colorectal tissues or the ovaries [1,2]. The cancer cells of PMP secrete copious amounts of gelatinous mucin in the peritoneal cavity, leading to nutritional compromise and death [3]. The current treatment for PMP involves rigorous cytoreductive surgery followed by hyperthermic intraperitoneal chemotherapy (HIPEC) [4,5]. This is a very invasive process with significant morbidity, but with 63% of patients with PMP survive beyond 10 years when this procedure is performed in specialized units [6]. When the colon is affected, the resection of the involved region with colon anastomosis is often performed [7]. During HIPEC, several different cytotoxic or cytostatic agents may be used, such as mitomycin C, doxorubicin, paclitaxel, cytostatics containing platinum, gemcitabine, docetaxel, melphalan, and irinotecan [8,9]; to increase the efficacy of the cytotoxics, an adjuvant comprising bromelain and acetylcysteine can be used [10].

Bromelain is an extract from the stem or the fruit of pineapple that contains numerous enzymes, such as cysteine protease, phosphatases, glucosidases, cellulases, and peroxidases [11]. The cysteine protease hydrolyzes peptide and glycoside bonds, which are prevalent in glycoproteins [12] in mucin secreted by PMP. Acetylcysteine, on the other hand, is an antioxidant that is capable of reducing disulfide bonds [13], which occur in abundance in PMP mucin. The use of these two agents in a combination known as BromAc^®^ has been shown to be a very effective mucolytic of pseudomyxoma peritonei (PMP) mucin [14,15], with anti-cancer effects on gastrointestinal (GI) cancer cells [16] and organoids [17]. BromAc^®^ has also been tested in the treatment of different intra-abdominal malignancies in vivo, such as colon cancer [18], pancreatic cancer [19], and a patient-derived xenograft (PDX) rat model of advanced appendiceal mucinous carcinoma peritonei (MCP) [20]. It has undergone phase I studies successfully [21], it is destined for phase II studies in PMP patients, and it is undergoing clinical assessment for use in the treatment of muco-obstructive respiratory diseases [22].

The healing process after colon anastomosis is rather complex and involves hemostasis, inflammation, proliferation, and remodeling phases [23]. Various proteases are secreted to degrade damaged tissues and thrombus and accelerate the repair process, whilst, at the same time, the immune system is activated in order to ensure that infections are controlled [24]. Bromelain has cysteine proteases that may aid the clearance of damaged tissues whilst enhancing the immune system [25]. Furthermore, it has been used for wound debridement and to enhance the healing process of burns and, currently, it is marketed under the trade name of Nexobrid^®^ [26]. Acetylcysteine has also been shown to enhance wound healing [27], and its mechanisms of action may be due to its antioxidant properties, which quench the reactive species abundant in injured tissues [13], whilst at the same time enhancing the immune system [28]. Further, it may also aid in the regeneration of endogenous antioxidants, such as glutathione [29]. Acetylcysteine is known to enhance the proteolytic action of bromelain and, since wound healing is dependent on debridement, amongst other parameters, such as immune enhancement, the modulation of inflammation, etc., the presence of acetylcysteine may, in fact, enhance the healing process [30]. Hence, as individual agents, bromelain and acetylcysteine are known to enhance wound healing. However, there are no studies to show their joint action on colon anastomosis healing. It is known that the failure of colon-anastomosis healing may result in anastomotic leakage, which is associated with serious complications and mortality [31].

Since wound healing depends on the deposition of collagen [32], the presence of bromelain and acetylcysteine may affect these two molecules. Collagen is composed of proteins with an abundance of peptide linkages and disulfide bonds [33] that may be affected by BromAc^®^.

Although bromelain and acetylcysteine appear to be relatively safe as individual agents in wound healing when applied externally, the delivery of these agents intraperitoneally following anastomosis has not been investigated before, nor has their combination in the form of BromAc^®^. Hence, we tested the effects of intraperitoneal BromAc^®^ on the healing process of colon anastomosis using a rat model. We initially measured the burst pressure of the anastomosis after colon anastomosis surgery to determine differences compared to control animals, which were only exposed to saline. Further, we examined the accumulation of collagen in both the groups at two time intervals, whilst, at the same time, determining the histology of the colon between the two groups pre- and post-treatment. Owing to the absorption of BromAc^®^ systemically with the exposure of different internal organs, we assessed the effect by comparing the histology of the tissues from the different organs from the two experimental groups in order to determine the histopathological effects.

## 2. Materials and Methods

### 2.1. Drug Preparation

BromAc^®^ was manufactured by Mucpharm Pty Ltd. (Kogarah, Australia) as a sterile solution. The drug diluent was 0.9% NaCl. The drug used in the study was prepared as one batch and stored at −30 °C until use.

### 2.2. Study Ethics and Design

The study protocol was reviewed and approved by UNSW Animal Care and Ethics Committee (ACEC), approval numbers 19/139A. After arrival at the BRC animal facility, 16 Wistar rats weighing an average of 250 g each were housed for at least one week for acclimatization, in adherence to standard protocol. The rats were divided into two groups (N = 8); control received intraperitoneal saline after anastomosis, whilst the other group received BromAc^®^ (experimental day 1). Half of the rats in each group (N = 4) were culled at 4 or 13 days post-surgery for assessment (Figure 1).

### 2.3. Colon-Anastomosis Surgery and Drug Treatment

All rats underwent laparotomy, colon anastomosis, and drug treatment on experimental day 1. Prior to all procedures, anesthesia was induced by 4% isoflurane inhalation. Anesthesia was confirmed using toe-pinch-response technique. Subsequently, anesthesia was maintained with isoflurane 1–3% inhalation. The surgical site was shaved, after which antiseptic skin chlorhexidine solution was applied. A 5 cm midline laparotomy was performed, and an anterior longitudinal 1.5 cm full-thickness colonic incision was made using scissors. Next, a colon-to-colon anastomosis was performed in a transverse fashion using interrupted 5/0 PDS absorbable stitches. Subsequently, two-layer closure of the abdomen was performed using running 3/0 PDS sutures. A thick local topical analgesic, lidocaine, was applied on the surgical site together with 0.01–0.05 mg/kg buprenorphine subcutaneously, every 6–12 hours, to ameliorate the pain. Both forms of analgesia (buprenorphine, topical analgesia) were applied if required, based on clinical signs of pain and distress. After the surgery, rats were injected with intraperitoneal BromAc^®^ (3/300 mg/kg) or sham-treated (9% Saline) using a sterile 25-gauge needle. The animals were then monitored until full recovery, with subsequent daily monitoring of their health and welfare parameters including activity, movement/gait, breathing, alertness, bodyweight, signs of intra-abdominal infection, anastomosis leak or rupture. At specified time intervals, 4 or 13 days post-surgery, rats from the control and treated groups were euthanized by inhalation of carbon dioxide (30% of chamber volume/min) and were then assessed for colon-anastomosis healing.

### 2.4. Anastomotic Burst Pressure

Post-euthanasia, the midline laparotomy was re-opened. Signs of inflammation or anastomotic leak were explored. Identification of anastomosis and recording of surrounding abscesses or adhesions were photographed. The colon was then ligated distal to the anastomosis above the peritoneal reflection. Colon was transacted 5 cm proximal to the anastomosis and a catheter was inserted into the colon, with fixation using 3/0 silk tie to prevent leak. The catheter was connected to a syringe and sphygmomanometer via a stopcock. The syringe was used to insufflate the colons gradually with air until a sudden loss of pressure occurred. This pressure was recorded as Anastomotic Burst Pressure (ABP). Colon anastomosis and visceral specimens were then collected for histological analysis.

### 2.5. Histological Evaluation

Formalin-fixed, paraffin-embedded sections of colon anastomosis and other viscera were prepared. Histopathological changes were assessed using H&E staining. Fibrosis was assessed based on trichrome-stained sections, with images captured using a binocular light microscope with a digital camera.

Histopathological features of colon post-anastomosis were evaluated using a semi-quantitative histopathology score. All tissues/slides were examined by an accredited contracted veterinary histopathologist for absence or presence of histopathological features. The pathologist was masked to group treatments but was familiar with background information. Histopathological features in the colon’s lamina propria were then segregated into two categories, at the surgical site and away from surgical site. The presence of histopathological findings was reported as number of animals which had the pathological features out of the 4 animals in each treatment group.

### 2.6. Statistical Analysis

Data were analyzed using GraphPad Prism version 9.0 (GraphPad Software, Inc., San Diego, CA, USA). Quantitative variables were compared using Student’s *t*-test and data were reported as the mean ± SD. Differences were considered statistically significant when *p* < 0.05.

## 3. Results

The autopsies performed on the control and treated animals revealed no anastomosis leaks or ruptures, and no infections. The body-weight measurements showed the same weight fluctuation pattern between the control and drug-treated groups (Figure 2A,B). However, the BromAc^®^-treated groups showed slightly greater weight loss during the first 5 days, with subsequent gains in weight, although these were not significant. The initial losses in weight in both the animal groups may have been due to surgical trauma. However, the animals in both the groups subsequently regained their weight (normal feeding). Other wellbeing parameters monitored during the course of the treatment showed normal health.

The healing of the colon anastomosis was assessed at two time periods in the animals (4 days & 13 days). All the colons were examined for the locations of burst perforations. The burst perforations were located at or next to the anastomosis sites. The outcomes of the anastomotic-burst-pressure experiment are shown in Figure 2C,D and Table 1. The t-test was used to compare the bursting-pressure values of the controls to those treated with the BromAc^®^, and there were no significant differences between the control and the BromAc^®^-treated groups at 4 days (*p* = 0.59) and 13-days (*p* = 0.10) post-surgery. Although this was not significant, on the 13th day, the BromAc^®^ group showed higher burst pressure. This was suggested by the slightly higher collagen density (again, the *t*-test was non-significant).

Inflammatory changes (lamina propria, submucosal, muscle layers, and serosal surfaces), the loss of structures/necrosis (e.g., epithelial cells, crypts), and vascular congestion, both at a point away from the site of the anastomosis and at the site of the anastomosis on H&E-stained sections were evaluated. No difference were between the control and the BromAc^®^-treated groups 4 days and 13 days post-surgery (Figure 3A; Table 2).

The fibrosis at the sites of colon anastomosis was interpreted based on trichrome-stained (collagen density) sections (Figure 3B). The quantitative analyses of the collagen density showed no differences in the percentage increase in collagen between the control and BromAc^®^-treated groups 4 days (*p* = 0.52) and 13 days (*p* = 0.26) post-surgery (Figure 3C). On the 13th day, there appeared to be a slightly higher collagen density, although the *t*-test did not show significance, owing to the small number of animals used in this study.

No abnormalities were detected upon the histopathological evaluations of the liver, kidney, spleen, pancreas, small intestine, heart, lung, trachea, and bronchus sections stained with H&E from the control and treated groups 4 and 13 days post-surgery (Figure 4). Hence, the results indicated the safety of the intraperitoneal concentration of BromAc^®^ administered over the study period.

## 4. Discussion

Colon anastomosis after the resection of diseased colon sections is a procedure that is carried out routinely on peritoneal cancer patients [7]. The subsequent healing process to re-establish a firm connection between the sections of the sutured colon involves a wound-repair process with the regeneration of tissues with increased tensile strength [23]. However, wound healing is a complex process that requires the presence of a number of chemokines with the enhanced formation of collagen, elastin, and fibronectin, the upregulation of immune cells, protein synthesis etc. [24].

Since bromelain contains cysteine protease and a number of other enzymes that primarily act on glycoside and peptide linkages with the hydrolysis of proteins and glycoproteins [12], and since it sis a fibrinolytic, it is envisaged that it may interfere with the healing process, since a number of protein molecules are involved in the reconstruction of new tissues. Similarly, acetylcysteine is a known reducing agent that may also affect the formation of tissues, although studies have indicated that acetylcysteine promotes wound healing [27]. Further, several studies show that bromelain has anti-inflammatory properties, which enhance wound healing; however, the effect of the combination of these two agents (BromAc^®^) on wound healing has not been investigated before. Dos Reis et al. observed a large reduction in chemokines and cytokines in the sputum of COVID-19 patients following BromAc^®^ treatment [22]. Patients with PMP have been treated with BromAc^®^; however, we did not investigate its effect on colon anastomosis [21]. The effect of BromAc^®^ delivered intraperitoneally is short-lived in the cavity owing to the absorption of the agents into the peritoneal membrane and, hence, their systemic distribution. However, BromAc^®^ has previously been delivered intra-tumorally into the mucinous mass, not post colonic anastomosis. Therefore, we proceeded to study how wound healing was affected in a rat model during colon anastomosis, along with the effect on the vital organs. Whilst there are potential benefits for healing associated with the components of BromAc^®^, the study team were very concerned about the possibility of anastomotic leakage due to the fibrinolytic and debriding nature of bromelain.

The present study on rats showed that there was a slight drop in body weights of the animals after the surgical interventions, indicating that the nutritional intake was reduced in both the control and the BromAc^®^-treated groups, suggesting slight trauma, pain, and discomfort in the animals. Noticeably, there was also a slightly greater loss of body weight in the BromAc^®^-treated group at experimental days 3–5, which eventually normalized. The loss of body weight was a general indication of wellbeing after surgical intervention in all the animals [34] (Figure 2A). From the seventh day to the 13th day, both the control and the BromAc^®^-treated group gained weight, indicating normal feeding habits or food intake (Figure 1B).

In our study, no anastomosis leakage was observed post-euthanasia in either the control or the BromAc^®^-treated rats. Previous studies using rat models showed very low incidences in control rats ranging, from 0–5% (0–1 out of 20 animals) [35,36]. A comparison of the burst pressure in the healing colon indicated no differences at 4 days after surgical intervention between the groups, and similar findings were observed at 13 days. Notably, the burst pressure at 4 days post-surgery was smaller than at 13 days (80 vs. about 120–150 mm Hg), indicating that at 4 days, the tissues were much weaker, with incomplete healing. Colon anastomosis was studied previously in Wistar rats. The recorded colon-burst pressure in the control rats varied due to differences in sex, the day of sacrifice post-operation, the type of suture, and the disease model [37,38,39,40,41]. However, overall, the results showed the increase in burst pressure with time, which was in conformity with our results [37,41]. Although, at 13 days, the differences between the control and the BromAc^®^-treated group were not significant according to the *t*-test, bromelain and acetylcysteine are known to enhance wound healing, which may have been the case in this study. However, owing to the small number of animals in the groups, we were unable to show any distinct differences.

Collagen is a fibrous protein that is specially assembled during wound healing at the site of an injury in order to strengthen the tensile quality of tissues, preventing rupture if pressure is applied. In the present work, the percentage collagen increases at 4 days shows similarities between the control and the BromAc^®^-treated groups (Figure 3C), with a 40% increase for both the groups at day four, indicating that the presence of BromAc^®^ does not interfere with collagen synthesis, which is highly important for wound healing.

Similarly, at 13 days, the collagen assessment again indicated no differences between the control and the BromAc^®^-treated rats, further demonstrating that BromAc^®^ does not interfere with collagen synthesis. The percentage collagen increase in the two groups (control and BromAc) at 13 days was slightly smaller when compared to the increase 4 days post- surgery, presumably due to the repair process reaching completion. Further, although not statistically significant (owing to the small number of animals in each group), there was a slightly greater amount of collagen in the BromAc^®^ group compared to control, which may indicate the enhancement of wound healing with increases in the amount of collagen synthesis.

Since the BromAc^®^ was administered intraperitoneally in these animals, the agent was absorbed by the peritoneal cavity and distributed systemically. Hence, we monitored the integrity of various other organs, such as the lungs, heart, kidneys, spleen, pancreas etc., by histology, in order to determine the end result of their pharmacological and biochemical effects on these vital organs. No abnormalities in the histological appearances of the various structures and cells in any of the vital organs were observed in either the saline- or the BromAc®-treated groups, indicating that at the concentration of BromAc^®^ delivered there was no effect on the microstructures of the internal organs, further indicating the complete systemic safety of the product at the concentrations in this protocol.

The present study indicates that colon-anastomosis healing is not affected by BromAc^®^ in healthy rats. Patients with PMP are often immunocompromised with other comorbidities; hence, there is a certain amount of uncertainty regarding the transfer of the current findings to patients. However, depending on the health of the PMP patient, as well as their age, nutritional status, and comorbidities, the healing process may vary, although the effect of BromAc^®^ on colon healing may be minimal or totally absent, in agreement with current animal studies. Furthermore, preclinical studies are required to ensure safety in the setting of an anastomosis, although current ongoing clinical studies without anastomosis have indicated no complications or concerns to date [42].

## 5. Conclusions

The results in this study suggest that the intraperitoneal delivery of the combination of bromelain and acetylcysteine (BromAc^®^) does not interfere with colon-anastomosis healing. The results also indicate the safety of the intraperitoneal delivery of BromAc^®^.

Although there are limitations in the current study, such as small number of animals in each group, the usage of animals that were robust and healthy, and the evaluation of colon anastomosis only, future studies may involve a more comprehensive approach, including the healing of other organs that may undergo resection during the surgical procedure. Further, additional studies should be designed to combine BromAc^®^ with chemotherapeutic agents and hyperthermia prior to expansion into early phase clinical trial.

## Figures and Tables

**Figure 1 cancers-15-03321-f001:**
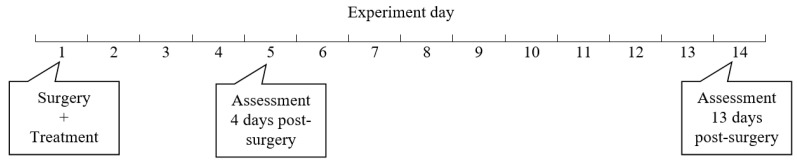
Research design. Sixteen Wistar rats were divided into two groups (N = 8); control received intraperitoneal saline after anastomosis, whilst the other group received BromAc^®^ (experimental day 1). Half of the rats in each group (N = 4) were culled for assessment at 4 or 13 days post-surgery (experimental days 5 and 14, respectively).

**Figure 2 cancers-15-03321-f002:**
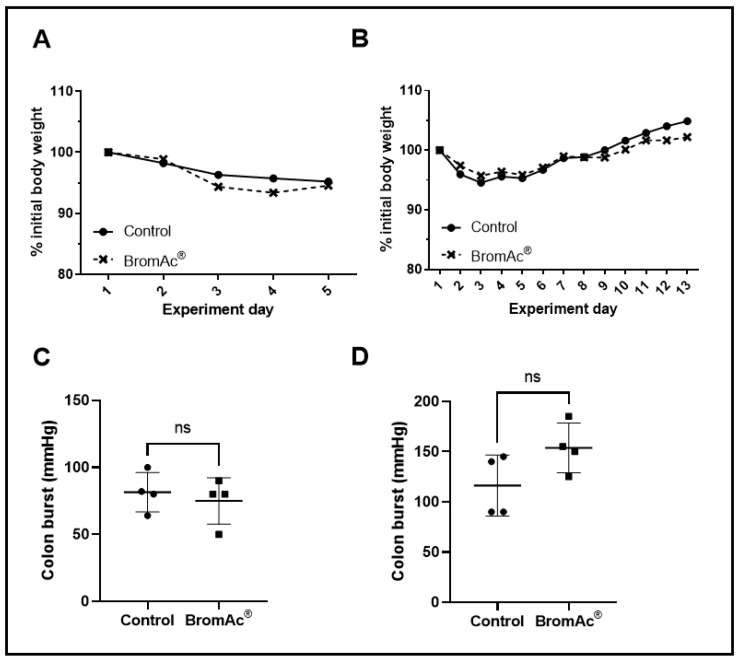
(**A**,**B**) Graphs show percentage changes in body weights of control and BromAc^®^-treated animals monitored for 4 and 13 days after colon-anastomosis surgery, respectively. (**C**,**D**) Graphs showing the potency of colon anastomotic burst pressure expressed by mmHg 4 days and 13 days post-colon-anastomosis surgery, respectively. Data presented as mean ± SD. Differences were considered statistically significant when *p* < 0.05, ns = not significant.

**Figure 3 cancers-15-03321-f003:**
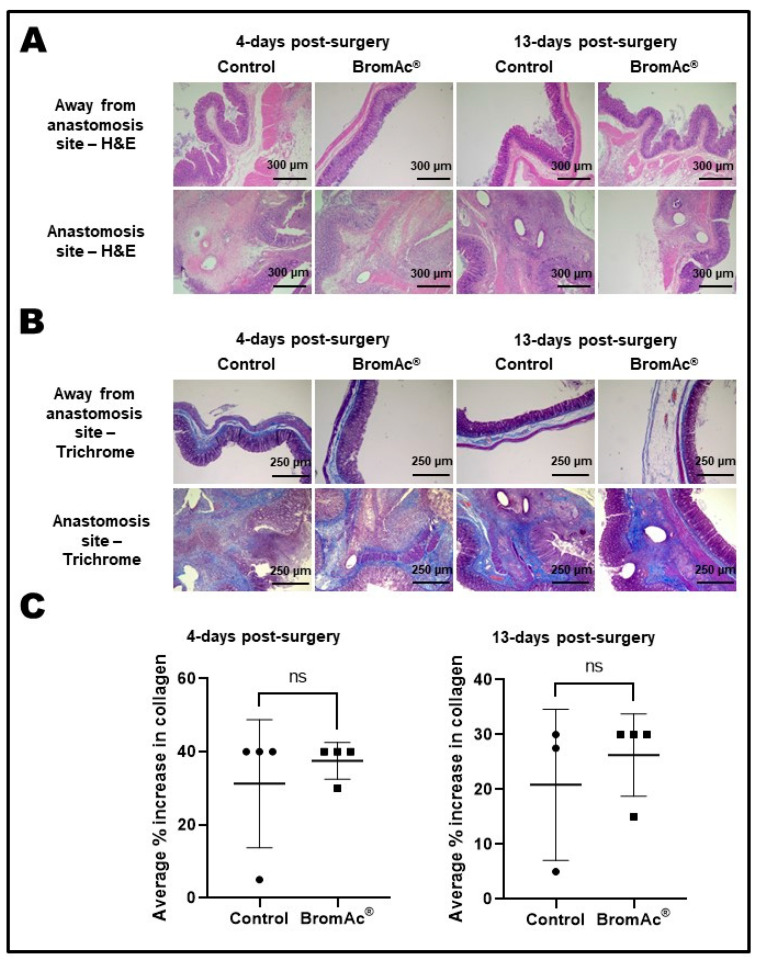
(**A**) H&E-stained histological images of colon resection and anastomosis sections. Final magnification, ×100; scale bar = 300 µm. (**B**) Trichrome-stained histological images of colon resection and anastomosis sections. Final magnification, ×50; scale bar = 250 µm. (**C**) Graphs show average percentage increase in collagen present in the colon-anastomosis sites. Data presented as mean ± SD. Differences were considered statistically significant when *p* < 0.05. ns = not significant.

**Figure 4 cancers-15-03321-f004:**
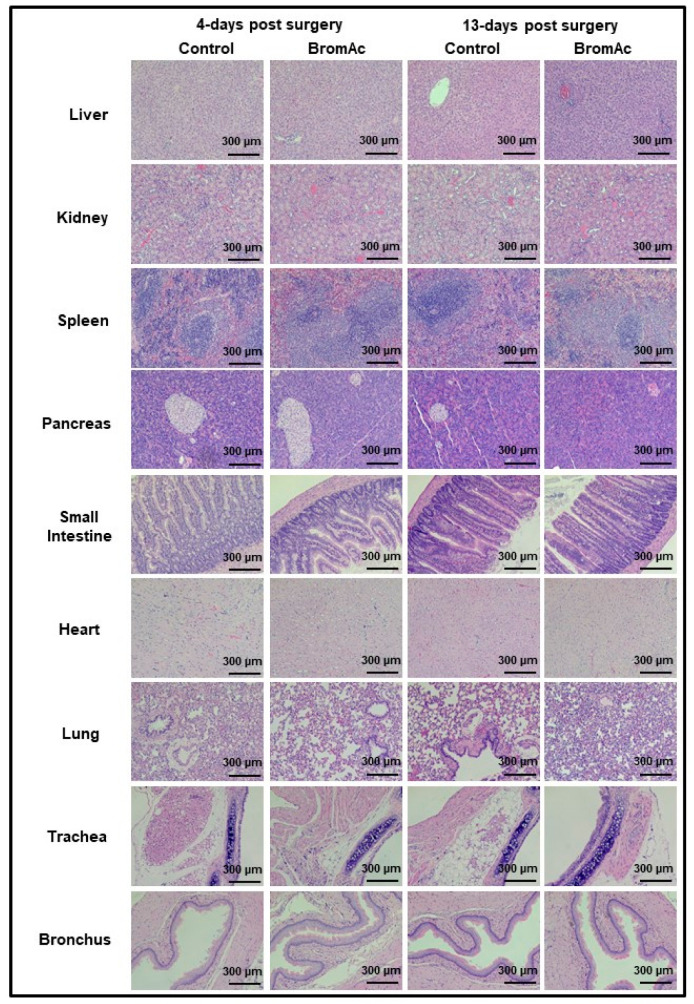
Microscopic histological images of liver, spleen, kidney, pancreas, and intestine stained with H&E. No abnormalities were detected upon histopathological evaluations when comparing BromAc^®^-treated group with sham control group 4- or 13-days post-surgery. Final magnification, ×100; scale bar = 300 µm.

**Table 1 cancers-15-03321-t001:** Bursting pressure values of the groups. Sixteen Wistar rats were either treated with intraperitoneal saline or BromAc^®^ post-colon-anastomosis surgery (N = 8/each treatment group). Half of the rats in each group (N = 4) were culled for assessment at 4 or 13 days post-surgery. Next, anastomotic burst pressure (ABP) test was performed to assess the tensile strength of colon healing. Data presented as range and (mean). Differences were considered statistically significant when *p* < 0.05.

4 Days Post-Surgery Range (Mean); mmHg			13 Days Post-Surgery Range (Mean); mmHg		
Control	BromAc^®^	*p*-Value	Control	BromAc^®^	*p*-Value
64–100 (81.50)	50–90 (75.00)	0.59	90–145 (116.3)	125–185 (153.8)	0.10

**Table 2 cancers-15-03321-t002:** Histopathological-features scores in t the rat-colon-anastomosis model. Rats were treated either with intraperitoneal BromAc^®^ or with saline. Histopathological features in the colon lamina propria were segregated into two categories, at the surgical site and away from the surgical site. The presence of histopathological findings was reported as the number of animals which had the pathological features out of the 4 animals in each treatment group. **^** Surgical site in one control animal could not be identified.

Histopathological Findings	4 Days Post-Surgery		13 Days Post-Surgery	
	Control ^	BromAc^®^	Control	BromAc^®^
**Colon, away from surgical site–lamina propria**				
Diffuse lymphocytic infiltration	3/3	2/4	0/4	0/4
Mixed inflammatory infiltration	0/3	2/4	4/4	4/4
**Colon, surgical site–lamina propria**				
Diffuse colitis, or mixed inflammation	4/4	3/4	4/4	4/4
Multifocal necrosis	4/4	4/4	1/4	1/4
Multifocal granulomata with multinucleated giant cells	0/4	0/4	3/4	4/4
Foreign-body pyogranuloma	0/4	1/4	2/4	1/4

## Data Availability

The datasets generated and/or analyzed during the current study are available from the corresponding author on reasonable request.

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
