# Peer review of "Intraperitoneal BromAc® Does Not Interfere with the Healing of Colon Anastomosis"

_cancers, 2023, doi:10.3390/cancers15133321_

Round 1

Reviewer 1 Report

Thank you for the opportunity to review this manuscript. The authors performed a rat experiment. If I'm right, 16 rats were divided between cases and controls. the authors compared the effect of Bromac vs saline on weight loss and gain after creating a colocolonic anastomosis and they investigated burst pressure.

The type of rats and the number must be included in the abstract.

In my view, the first clinical application for Bromac would be percutaneous infucion in symptomatic mucine collections. I understand that the authors would like to know whether Bromac intraperitoneally is safe. But they already tested this in humans. With the aforementioned indication, I do not see the additional value to test this in a rat model with a bowel anatomosis.

Non statistically significant differences should not be highlighted as they represent no difference based on your chosen P value.

In the discussion, the authors should adequately discuss the translation of their findings to patients. If the anastomosis heals well in a rat, it does not mean that the same is true in an often sick patient.

Author Response

Dear Reviewer,

Thank you for reviewing our manuscript. Much appreciated.

Here, we responded to your questions:

  1. The type of rats and the number must be included in the abstract.

The number and type of rats are included in the abstract. Due to the maximum number of abstract words (200 words), we had to change the abstract slightly.

  1. In my view, the first clinical application for BromAc would be percutaneous infusion in symptomatic mucin collections. I understand that the authors would like to know whether BromAc intraperitoneally is safe. But they have already tested this in humans. With the aforementioned indication, I do not see the additional value to test this in a rat model with a bowel anastomosis.

Thank you for the question. Yes, patients have been treated with BromAc, however, we did not investigate the effect on colon anastomosis. The effect of BromAc delivered intraperitoneally is short-lived in the cavity owing to the absorption of the agents into the peritoneal membrane and hence systemically distributed. However, BromAc is delivered intra-tumoral into the mucinous mass. Hence, after mucolysis, BromAc is readily absorbed systemically via the peritoneal membrane with a residual amount absorbed by the digestive and other peritoneal organs. This explanation is now included in the discussion section.

  1. Non-statistically significant differences should not be highlighted as they represent no difference based on your chosen P value.

The non-significant effect on wound healing is now un-highlighted in the abstract. Thank you.

  1. In the discussion, the authors should adequately discuss the translation of their findings to patients. If the anastomosis heals well in a rat, it does not mean that the same is true in an often-sick patient.

Well, no interference of healing of colon anastomosis in rats by BromAc in healthy rats does not translate to sick and immunocompromised patients, that is correct, however, it is just evidence and indicates reasonable safety with colon anastomosis healing. Cancer patients with colon anastomosis healing will typically vary with the health of the patient. However, our aim is to show that BromAc does not interfere with wound healing and using sick rats as a model may complicate the study. This part is now included in the discussion, thank you.

Reviewer 2 Report

I have read with interest this study from David Morris' team. BromAc treatment is in evaluation right now with a only one phase I trial outside Australia performing by Spanish group (preliminary results could be cited: 10.1016/j.ejso.2021.12.296 )   with promising results the both phase I presented.   One of the important questions about the use of BromAc is its effect on colon anastomosis. This study could report some evidence about the safety of BromAC on colon anastomosis healing. 

However I have some comments about the study: 

·      What is the rate of colon anastomosis leakage in Wistars? Why the authors have chosen 4 rats per groups? it is must be justified since no leakage was seen in any group maybe the sample size is too small. 

·      Why was the dose 3mg/Kg instead of the dose used in the phase I trial?

 "Drug doses were determined based on estimated tumour volume (Brom 30 mg NAC 1·5 g tumour <250 ml, Brom 45 mg NAC 1·5 g tumour >250 ml or multiple treatment sites, intraperitoneal Brom 45 mg NAC 1·5 g tumour burden low or Brom 60 mg NAC 2 g extensive tumour. In patients that had more than one drain placed, the drug was administered between the sites based on the volume of tumour at each. The maximum daily dose of Brom and Ac were 60 mg and 2 g, respectively" Phase I trial from Australia

“0.5mg per mL of total tumor volume calculated, administered percutaneously” phase I from Spain  https://clinicaltrials.gov/ct2/show/NCT04982146

·      In Methods section, the method to analyze the inflammation or signs of anastomotic leakage, must be described and not in Results section.

·      For statistical analysis the qualitative variables cannot be compared by Student’s t test, only for quantitatives.

·      There are some preliminary studies about the normal pressure for colon anastomosis in Wistar? It must be reported as control pressure and discussed it.

·      Table 1 needs the n size for each group

·      Authors have evaluated the different types of collagen and their distribution after BromAC treatment?

·      In the page 7 line 218 there is a table without any reference and without footnote.

·      In my opinion the figure 4 is not needed for the objectives of this study.

·      The conclusions cannot say that this study shows, it must be changed by “suggests”.  

Author Response

Dear reviewer,

Thank you for reviewing our manuscript. Much appreciated.

Here we responded to your questions:

  • What is the rate of colon anastomosis leakage in Wistars? Why have the authors chosen 4 rats per groups? it is must be justified since no leakage was seen in any group maybe the sample size is too small.

Previous investigators reported very low incident of leakage in control rats. Ghiselli et al 2020 reported 1 out of 20; 5%) leakage in control. Despoudi et al 2021 reported no leakage in control (0 out of 20). This has been included in the discussion. The present experiment uses small number of animals (8 in each group (4 + 4)-control and (8 – (4 + 4) in treatment group owing to the time consuming and nature of the procedures. Besides, 4 in each group gives sufficient statistical power although a larger number would give a significant statistical power. Experiments such as this need sufficient surgical skills (theatrical surgeon) and we have used such skills and hence forth although the number of animals is small, it is sufficiently strong.

  • Why was the dose 3mg/Kg instead of the dose used in the phase I trial?

 "Drug doses were determined based on estimated tumour volume (Brom 30 mg NAC 1·5 g tumour <250 ml, Brom 45 mg NAC 1·5 g tumour >250 ml or multiple treatment sites, intraperitoneal Brom 45 mg NAC 1·5 g tumour burden low or Brom 60 mg NAC 2 g extensive tumour. In patients that had more than one drain placed, the drug was administered between the sites based on the volume of tumour at each. The maximum daily dose of Brom and Ac were 60 mg and 2 g, respectively" Phase I trial from Australia

“0.5mg per mL of total tumor volume calculated, administered percutaneously” phase I from Spain https://clinicaltrials.gov/ct2/show/NCT04982146

This dose is used here as a sensitizer not as mucolytic agent. In addition, this dose has been used IP in previous animal studies and has been shown to be safe and effective.

Mekkawy, A. H., Pillai, K., Badar, S., Akhter, J., Ke, K., Valle, S. J., & Morris, D. L. (2021). Addition of bromelain and acetylcysteine to gemcitabine potentiates tumor inhibition in vivo in human colon cancer cell line LS174T. American Journal of Cancer Research11(5), 2252.

Mekkawy, A. H., Pillai, K., Suh, H., Badar, S., Akhter, J., Képénékian, V., ... & Morris, D. L. (2021). Bromelain and acetylcysteine (BromAc®) alone and in combination with gemcitabine inhibit subcutaneous deposits of pancreatic cancer after intraperitoneal injection. American Journal of Translational Research13(12), 13524.

  • In Methods section, the method to analyze the inflammation or signs of anastomotic leakage, must be described and not in Results section.

The method to analyze inflammation or signs of anastomotic leakage has been included in the correct section.

  • For statistical analysis the qualitative variables cannot be compared by Student’s t test, only for quantitative.

The “qualitative” word has been corrected to “quantitative”, thank you.

  • There are some preliminary studies about the normal pressure for colon anastomosis in Wistar? It must be reported as control pressure and discussed it.

This has been discussed in the discussion section “Colon anastomosis has been studied previously in Wistar rats. Recorded colon burst pressure in control rats is varied due to variance in sex, day of scarification post-operative, type of suture, and disease model [34-38]. However, collective results showed the increase of burst pressure with time which is in conformity with our results [34, 38]”

  • Table 1 needs the n size for each group.

Table 1 with the number of rats in each group (n value) has been included.

  • Authors have evaluated the different types of collagen and their distribution after BromAC treatment?

We have used trichrome to stain general collagen fibres.

  • In the page 7 line 218 there is a table without any reference and without footnote.

Corrected, it should be table 2.

  • In my opinion figure 4 is not needed for the objectives of this study.

Figure 4 in the present script denotes histological similarities between treated and control as BromAc is absorbed systemically via the peritoneal membrane with a residual amount absorbed by the digestive and other peritoneal organs.

  • The conclusions cannot say that this study shows, it must be changed by “suggests”.

“Show” changed into “suggest” in the conclusion section, thank you.

Reviewer 3 Report

Re: Intraperitoneal BromAc® does not interfere with the healing of colon anastomosis.

The authors provide a reasonable experimental schema for evaluation the impact of their combined mucolytic agent on intraabdominal anastomosis. The question and the experimental plan is straightforward and the authors provide reasonable histologic analyses examining both mechanical and histologic factors implicated in anastomotic leaks. However the work is missing critical points that are necessary in a scientific manuscript and there are some major points that need revision prior to publication.  

1.    Pseudomyxoma peritonei is a phenomena that occurs as a sequalae of disease processes such as appendiceal, colorectal and gynecologic cancers. PMP in itself is not a disease. This should be corrected on the simple summary and abstract. There is also incongruency between the type of bowel connections in the simple summary and the abstract: bowel addresses all bowel, small bowel and colon are different. They have different tensile strength. Colon was likely selected as a model given that colon has lower tensile strength and higher risk of anastomotic leak. But this should be clarified as one simple summary addresses bowel and the abstract addresses colon.

2.    Introduction: The introduction statements on BromAc seem to imply that the agents in BromAc may be advantageous to wound healing rather than a liability. The data supports neither – which is a valuable piece of information when dissected down using an animal model as this is something that we cannot reliably and physiologically study in humans. Would revise paragraph 3 of introduction and paragraph 2 of discussion (first two statements) to better indicate the scientific uncertainty.

3.    Materials and methods: 16 mice are used total in the experiment, but it is unclear the breakdown ratio between control and experimental groups. Please clearly describe if this was 4 in each of the 4 groups/time points?

4.    Colon burst pressure test: How rapidly was the colon inflated over time? Where was the location of the colonic perforation in response to the pressure? Was it reliably at the anastomosis or elsewhere? This would be important information to convey.

5.    Other factors important in healing of bowel anastomosis: The study primarily focuses on burst pressure, collagen density, and histological evaluation. In humans, reasons for anastomotic leak are multi-factorial. Other additional parameters, such as inflammation markers or other relevant physiological indicators should be considered to provide a more comprehensive assessment of the healing process. The authors begin to broach this by stating that they evaluate inflammatory changes but there is no mention as to how this is done: was this evaluation and scoring by one or multiple pathologists with expertise in veterinary pathology? What is the scoring system?

6.    Table 2 is labeled as figure 2 in the manuscript.

7.    Discussion: While the limitations of small sample size is noted, there are other points that would be valuable for discussion to put the current findings into richer context. Discussion such as rationale for selection of experimental approaches to testing anastomotic integrity. The authors are to be commended for selecting both an approach that tests both tensile mechanical strength and performing histologic correlates, but where are other tests and why were these selected? How can these results begin to be extrapolated to human testing?

8.    The authors mention that Bromac is being used in clinical studies; are there any available data on bowel anastomoses in these early studies? If so what are the results?

9.    Limited conclusion: The conclusion is too brief and does not adequately summarize the key findings or their implications and its conclusions of safety need to be justified of the results in pre-clinical models. For example the authors describe "complete systemic safety" which is not necessarily true; there appeared to be safety based on the tests the ran. The authors could provide a more comprehensive conclusion, discussing the overall significance of their results, potential limitations, and future directions for research.

Quality of English was adequate to convey the major points. Minor stylistic issues such as the selection of the European preposition/conjunction "whilst" or other sentence structures do not affect the quality of the data.

Author Response

Dear reviewer,

Thank you for reviewing our manuscript. Much appreciated.

Here we responded to your questions:

  1. Pseudomyxoma peritonei is a phenomena that occurs as a sequalae of disease processes such as appendiceal, colorectal and gynecologic cancers. PMP in itself is not a disease. This should be corrected on the simple summary and abstract. There is also incongruency between the type of bowel connections in the simple summary and the abstract: bowel addresses all bowel, small bowel and colon are different. They have different tensile strength. Colon was likely selected as a model given that colon has lower tensile strength and higher risk of anastomotic leak. But this should be clarified as one simple summary addresses bowel and the abstract addresses colon.

In the simple summary, the word disease has been removed and the word bowel has been replaced with the word colon.

  1. Introduction: The introduction statements on BromAc seem to imply that the agents in BromAc may be advantageous to wound healing rather than a liability. The data supports neither – which is a valuable piece of information when dissected down using an animal model as this is something that we cannot reliably and physiologically study in humans. Would revise paragraph 3 of introduction and paragraph 2 of discussion (first two statements) to better indicate the scientific uncertainty.

We have revised para. 3 in the introduction and para 3 in the discussion to indicate the scientific uncertainty of the combination of bromelain and acetylcysteine in wound healing.

  1. Materials and methods: 16 mice are used in total in the experiment, but it is unclear the breakdown ratio between control and experimental groups. Please clearly describe if this was 4 in each of the 4 groups/time points?

The breakdown of 16 rats in 4 groups has been clearly indicated.

  1. Colon burst pressure test: How rapidly was the colon inflated over time? Where was the location of the colonic perforation in response to the pressure? Was it reliably at the anastomosis or elsewhere? This would be important information to convey.

The syringe was used to insufflate the colons gradually with air until a sudden loss of pressure occurred. All colons have been examined for the location of burst perforations. The burst perforations were located at or next to anastomosis sites. This has been added to the methods and results sections, respectively. Thank you. 

  1. Other factors important in healing of bowel anastomosis: The study primarily focuses on burst pressure, collagen density, and histological evaluation. In humans, reasons for anastomotic leak are multi-factorial. Other additional parameters, such as inflammation markers or other relevant physiological indicators should be considered to provide a more comprehensive assessment of the healing process. The authors begin to broach this by stating that they evaluate inflammatory changes but there is no mention as to how this is done: was this evaluation and scoring by one or multiple pathologists with expertise in veterinary pathology? What is the scoring system?

Thank you. This part is now included in the methods section “Histopathological features of colon post-anastomosis were evaluated using a semi-quantitative, histopathology score. All tissues/slides were examined by an accredited contracted veterinary histopathologist for absence or presence of histopathological features. The pathologist was masked to group treatments but was familiar with background information. Histopathological features in the colon’s lamina propria were then segregated into two categories, at the surgical site and away from the surgical site. The presence of histopathological findings was reported as the number of animals which have the pathological feature out of the 4 animals in each treatment group.”

  1. Table 2 is labeled as figure 2 in the manuscript.

Corrected, thank you.

  1. Discussion: While the limitations of small sample size is noted, there are other points that would be valuable for discussion to put the current findings into richer context. Discussion such as rationale for selection of experimental approaches to testing anastomotic integrity. The authors are to be commended for selecting both an approach that tests both tensile mechanical strength and performing histologic correlates, but where are other tests and why were these selected? How can these results begin to be extrapolated to human testing?

These specific tests were selected mainly because they relate directly to wound healing in colons. The research approach has been included in the end of introduction:

“Although bromelain and acetylcysteine appear to be relatively safe as individual agents in wound healing when applied externally, the delivery of these agents intraperitoneally has not been investigated before, neither the combination as BromAc®. Hence, we tested the effects of intraperitoneal BromAc® on the healing process of colon anastomosis using a rat model. We initially measured the burst pressure of the anastomosis after colon anastomosis surgery to determine any difference compared to control animals that were only exposed to saline. Further we examined the accumulation of collagen in both the groups at two-time intervals whilst at the same time determining the histology of the colon between the two groups pre and post treatment. Owing to absorption of BromAc® systemically with exposure of different internal organs, we assessed the effect by comparing the histology of tissues from the different organs of the two experimental groups in order to determine any histopathological effect.

The discussion has been updated to include the translation of this research findings into human:

“The present study indicates that colon anastomosis healing is not affected by BromAc®, in healthy rats, PMP patients are often immunocompromised with other comorbidities and hence there is a certain amount of uncertainty in transferring the current findings to patients. However, depending on the health of the PMP patient, their age, nutrition and comorbidities, the healing process may vary although the effect of BromAc® on colon’s healing may be minimal or totally absent in agreement with current animal studies. Besides current ongoing clinical studies have indicated no complication or concerns.”

  1. The authors mention that Bromac is being used in clinical studies; are there any available data on bowel anastomoses in these early studies? If so, what are the results?

Thank you for the question. Yes, patients have been treated with BromAc, however we did not investigate the effect on colon anastomosis. The effect of BromAc delivered intraperitoneally is short lived in the cavity owing to absorption of the agents into peritoneal membrane and hence systemically distributed. However, BromAc is delivered intra tumoral into the mucinous mass and hence after mucolysis, BromAc is readily absorbed systemically via the peritoneal membrane with residual amount absorbed by the digestive and other peritoneal organs. This explanation is now included in the discussion section.

  1. Limited conclusion: The conclusion is too brief and does not adequately summarize the key findings or their implications and its conclusions of safety need to be justified of the results in pre-clinical models. For example, the authors describe "complete systemic safety" which is not necessarily true; there appeared to be safety based on the tests the ran. The authors could provide a more comprehensive conclusion, discussing the overall significance of their results, potential limitations, and future directions for research.

We have discussed further the limitations of this study in both the discussion and conclusion sections.

Round 2

Reviewer 2 Report

The authors have corrected all my requests. It could be suitable for publication